# Children's outdoor play at early learning and child care centres: Examining the impact of environmental play features on children's play behaviour

Rachel Ramsden[1,2,3]*, Ian Pike[1,2,4], Sally Thorne[5], Mariana Brussoni[1,2,3,6]

1 Department of Pediatrics, University of British Columbia, Vancouver, British Columbia, Canada,
2 British Columbia Children's Hospital Research Institute, Vancouver, British Columbia, Canada, 3 School of Population & Public Health, University of British Columbia, Vancouver, British Columbia, Canada,
4 British Columbia Injury Research & Prevention Unit, Vancouver, British Columbia, Canada, 5 School of Nursing, University of British Columbia, Vancouver, British Columbia, Canada, 6 Human Early Learning Partnership, University of British Columbia, Vancouver, British Columbia, Canada

* rramsden@bcchr.ca

## Abstract

Early learning and child care centres are critical settings to support children's regular, repeated and quality time spent in outdoor play. Gibson's theory of affordances highlights the importance of the human-environment relationship, emphasizing how children use environmental information to inform their behaviour. This study aims to understand the association between children's outdoor play behaviour and common environmental play features in early learning and child care outdoor play spaces, through the behaviour patterns of children. Children's play behaviour was collected via observational behaviour mapping at eight early learning and child care centres in the Greater Vancouver region between September 2021 and November 2022, as part of the PROmoting Early Childhood Outside study. A multivariate logistic regression model examined the association between outdoor play behaviour and environmental play features, via odds ratio and 95% confidence intervals. The results indicate environmental play features, including gardening areas, playhouses, climbing structures and tricycle paths supported increased opportunities for children's outdoor play. Gardening areas, playhouses, sandboxes, outdoor stages and fixed water features provided opportunities for exploratory play, while climbing structures and trike paths provided opportunities for physical play. Opportunities for diverse forms of play were less realized in dedicated open play areas, with the availability of loose parts and moveable equipment primarily influencing these spaces. The results of this study have important implications for future early learning and child care outdoor space design. Further research should consider children's dynamic movement and transition between outdoor affordances, and the influence of loose parts on the use of environmental play features.

**Data availability statement:** All data and relevant materials are available in the Borealis UBC Research Data Collection at https://bore-alisdata.ca/dataverse/UBC_RD (DOI: 10.5683/SP3/WGHKXC).

**Funding:** This research was funded by the Lawson Foundation, grant GRT 2020-137. The funders had no role in study design, data collection and analysis, decision to publish, or preparation of the manuscript.

**Competing interests:** The authors have declared that no competing interests exist.

## Introduction

Early learning and child care (ELCC) environments are a fundamental component of a child's microsystem [1]. Outside of the home, ELCC settings are the most influential environment in a child's life for outdoor play to transpire [2,3]. Research has shown that outdoor play in the early years is associated with many health and developmental benefits, including enhanced cognitive, physical, emotional and social development, spatial awareness, motor-skills, and physical activity [4–8]. Despite these benefits, contemporary trends, such as changing neighbourhood landscapes, increased screen time, and the prevalence of structured activities, combined with families' longer work hours, have led to a reduction in the time children spend engaged in outdoor play [9,10]. Shortcomings in sufficient outdoor play participation are also visible within the ELCC landscape; research demonstrates that children routinely do not receive adequate outdoor play opportunities while attending structured settings, such as ELCC centres and primary schools [11,12].

The provision of outdoor play in ELCC settings is influenced by a range of factors, including geographic location, seasonality, educator training and retention, family education, centre-level policies, government legislation and regulations, and the physical built environment [13]. Physical environment correlates, such as play space size, number of distinct play areas, and presence of natural elements and loose parts have been previously identified as critical predictors of children's outdoor play in ELCC environments [13,14]. Larger outdoor play spaces enhance children's play duration and physical activity participation, particularly when spaces are beyond the minimum requirements established by government licensing or regulatory frameworks [14,15]. Additionally, the presence of multiple, distinct play zones within an ELCC centre's outdoor space and the inclusion of naturalized environments have been shown to promote greater frequency and diversity of outdoor play experiences among children. [13,16]. Thoughtful combinations of natural and built environment elements, as well as purposeful circulation between outdoor play features, can enhance children's outdoor play and support a greater use of the space [17,18]. The presence of loose parts supports higher levels of play engagement and more diverse activities than environments featuring only fixed equipment or purpose-defined materials [19,20]. Although considerable evidence underscores the influence of physical environmental factors on children's outdoor play in ELCC centres, there remains a limited understanding of the play value offered by specific fixed play features and designated areas within these outdoor settings.

Similar to conventional playground design, outdoor play spaces in ELCC centres frequently incorporate a consistent set of elements shaped by considerations of convenience, design efficiency, and the availability of resources [17]. Often referred to as "traditional," "modular," or "kit of parts" designs [17,21], ELCC environments typically include standardized fixed features such as climbing structures, sandboxes, and gross motor areas [17,22–24]. The consistent use of these features is influenced by their ease of installation and upkeep, as well as regulatory guidelines, financial limitations, and perceived assumptions about their developmental value [25]. Some studies have sought to understand the influence of these commonly found play elements

and areas on children's behaviour, focusing on physical activity, movement patterns, and sedentary behaviour [23,26,27]. Open spaces and tricycle pathways have been positively associated with children's moderate-to-vigorous activity intensity [23,27,28], while conflicting evidence exists on the influence of fixed climbing structures [23, 26, 28, 29], sandboxes [23,26,28] and gardening areas [29,30] on children's physical activity levels. Researchers often interpret such findings in relation to children's outdoor play, however, outdoor play comprises a broader range of behaviours, many of which do not involve moderate or vigorous movements. As the connection between the physical outdoor environment and children's play engagement gains recognition, it is crucial to explicitly examine children's diverse play behaviours and consider them as distinct from physical activity.

Children's outdoor play encompasses a diverse range of skills, activities and engagement, rather than the presence or absence of one defining characteristic [31]. Outdoor play can include functional play [32,33], constructive play [32,33], social play [34] and dramatic play [35], as well as structured games [36]. Therefore, outdoor play spaces should consider opportunities for diverse play forms, not solely focusing on promoting physical activity or a single form of play to transpire [16]. Refshauge et al. [37] proposed that environmental opportunities for play should be climb-able, jump-on-able, run-able, balance-able, sing-on-able, imagine-able, touch-able, move-able, mould-able and construction-able. When children are offered more instances for diverse forms of play to transpire, the play value of an environment is enhanced [38,39]. To support diverse outdoor play opportunities, the equipment and materials present in outdoor spaces should encourage a wide range of interactions and forms of play to transpire [40]. Existing literature on children's outdoor play and the relationship with ELCC environments often overlooks the complex, multifaceted nature of play and its broader developmental benefits. To develop a more comprehensive understanding of how fixed outdoor elements and designated play areas influence children's outdoor play, it is essential to consider both the measurable aspects of movement and the experiential qualities that shape play.

## Theory of affordances

One approach to exploring the relationship between the environment and children's outdoor play is to examine the affordances offered by a space. Gibson's theory of affordances posits that the physical environment offers various action possibilities, or affordances, that are perceived directly by individuals based on their capabilities and the properties of the environment [41]. Affordances represent opportunities for children's action, both perceived or actual, within their outdoor environment. This theoretical framework considers the person-environment relationship as transactional, reciprocal and based on human perception [41,42]. For children, affordances are dynamic and context-dependent; an element of the environment can change for a child depending on the situation, abilities or goals [43,44]. In addition, affordances can hold a different meaning and potential for each child based on the child's knowledge, experience, strength, size, skills and preferences [43]. In the context of children's outdoor play, a slide may serve as a sliding apparatus for some and a climbing structure for others. Alternatively, on a rainy day, a slide may become a water race track for rocks, leaves and other loose parts. This relational view perspective between the physical environment and children's actions provides a lens for analyzing and theorizing how specific features of outdoor environments influence the play behaviours of children [45].

## Research purpose

The predictive influence of outdoor environments in shaping children's play behaviour has led to an increasing demand for research focused on identifying optimal design strategies to effectively support and enrich children's play experiences. This study conceptualizes outdoor play as distinct from physical movement and encompassing multiple and diverse forms of play. While existing research has examined the outdoor physical environment in ELCC settings, primarily related to physical activity outcomes, there remains limited understanding of the specific affordances for children's outdoor play offered by fixed environmental features and areas. The objective of this research is to analyze the association between outdoor play features commonly found in urban ELCC centres and children's outdoor play behaviour.

## Materials and methods

### Umbrella study

This study is part of the PROmoting Early Childhood Outside (PRO-ECO) randomized controlled trial that aims to increase the amount of time children spend participating in outdoor play in ELCC centres. The PRO-ECO study evaluates the PRO-ECO intervention, which includes a built environment modification to outdoor play spaces, within eight ELCC centres in the Greater Vancouver region (Canada). The ELCC centres involved in the PRO-ECO study were selected based on specific criteria: they were managed by the same organization (YMCA), served children aged 2.5 to 6 years, participated in the Affordable Child Care Benefit, were located near researchers at the [redacted for anonymous review], and were considered ready to join the study following informal interviews with the research team. The PRO-ECO trial collected data from September 2021 to November 2022 at three time points: Fall 2021 (Time 1), Spring 2022 (Time 2), and Fall 2022 (Time 3). For this study, PRO-ECO data across all time points were pooled to create a cross-sectional dataset and maximize observation points for this study. A detailed study protocol for the PRO-ECO study was published elsewhere [46].

### Inclusion/ exclusion criteria

Eight ELCC centres delivering full-day group child care and participating in the PRO-ECO trial were included in this study. Children in this study were aged 2–6 years and attending a participating ELCC centre. Continuous recruitment of children occurred between August 2021 and November 2022 through ECE's at each ELCC centre who distributed consent forms and initiated face-to-face conversations with families. Over the course of the study, a total of 217 children participated in the PRO-ECO trial and are included in this study's dataset. Approval for this study was granted by the [redacted for anonymous review]. Written parental consent was obtained for all children involved in the study.

### Study sites

The eight participating centres were located in 3 different urban cities within the Greater Vancouver region and operated by the YMCA. Participating ELCC centres were similar in size, topography, available play features and pedagogical approaches, such as the inclusion of loose parts, due to their operation by the same organization and comparable geographic location. All participating centres had outdoor spaces that were directly adjacent to their indoor space. Three of the ELCC centres were located above-grade and had rooftop outdoor play spaces. The remaining five centres had indoor and outdoor spaces located at-grade. Most of the participating centres had outdoor spaces between 170–230 m$^2$, however, two centres had larger outdoor spaces (Centre B & Centre C). The eight ELCC centres were topographically flat, with minimal uneven or steep surfacing. All centres had a climbing structure, with most containing a traditional fabricated play structure with a climbing wall, stairs and a slide. Concrete surfacing was present in all eight participating centres and was often used as a gross motor tricycle path. Each of the participating centres had gardening areas, including raised concrete planters and at-grade garden beds. Further information on the eight participating centres can be found in Table 1 and Fig 1.

### Data collection

This study used observational behaviour mapping (OBM) to measure children's outdoor play behaviour in association with their outdoor ELCC space during dedicated outdoor times. This approach strives to understand how an environment supports movement behaviours by mapping, recording, organizing, displaying, and analyzing geographically located data [47,48]. OBM completes recordings of participant movement and behaviours within a given environment to determine how participants use a designated space [48]. In this study, OBM was used to collect approximately 200, 15-second observational behaviour points at each ELCC centre and data collection time point. Data were collected over 30 days at each

**Table 1. Characteristics of participating ELCC centres (n = 8).**

| Attribute | Centre | | | | | | | |
|---|---|---|---|---|---|---|---|---|
| | Centre A | Centre B | Centre C | Centre D | Centre E | Centre F | Centre G | Centre H |
| Approximate size (m²) | 335 | 171[a] | 754 | 196 | 222 | 270 | 173 | 207 |
| Grade[b] | Above-grade | Above-grade | At-grade | At-grade | Above-grade | At-grade | At-grade | At-grade |
| Climbing structure | Play structure with slide, wood stumps | Play structure with slide | Climbing hill with slide, boulders, intertwined climbing logs* | Play structure with slide, boulders, wood stumps | Play structure with slide | Play structure with slide, balance logs, wood stumps | Play structure with slide, boulders | Wood cubes, balance logs, wood stumps |
| Gross motor path | Concrete tricycle path | Concrete tricycle path | Concrete tricycle path | Concrete tricycle path | None | Concrete tricycle path | Concrete tricycle path | Concrete tricycle path; gravel path |
| Surfacing materials | Concrete, natural soil/ dirt, mulch, artificial turf, wood decking* | Concrete, rubber, wood decking | Concrete, natural soil/ dirt, mulch | Concrete, natural soil/ dirt, mulch, rocks | Concrete, rubber, wood decking | Concrete, mulch, natural soil/ dirt | Concrete, mulch, natural soil/ dirt | Concrete, mulch, natural soil/ dirt, gravel |
| Gardening area | Raised concrete planters | Raised concrete planters | Garden beds | Garden beds, raised concrete planters | Raised concrete planters | Garden beds, raised concrete planters | Garden beds, raised concrete planters | Garden beds, raised concrete planters |
| Water feature | None | None | None | Rain catchers* | None | None | Water pump (moveable)* | Water pump & trough (fixed) |
| Sandbox | Yes | Yes | Yes | Yes | No | Yes | Yes | Yes |
| Mud kitchen | No | Yes | No | No | No | No | No | No |
| Table area | No | Yes | Yes | No | Yes | No | No | Yes |

*These items were modified/added as part of the PRO-ECO intervention before Time 2 data collection.

[a]This centre increased in size minimally at Time 3 due to an extension to their outdoor space as part of the PRO-ECO intervention.

[b]Grade refers to the ground relationship to the building. Above grade indicates an outdoor play space above ground level. At-grade indicates an outdoor play space at ground level.

timepoint: October 2021 – December 2021 (Time 1); April 2022 – June 2022 (Time 2); October 2022 – December 2022 (Time 3). These data collection periods were selected as part of the PRO-ECO study protocol [46] and strived to maintain seasonal similarities between all data collection timepoints.

Within the OBM protocol [46], data were collected on children's outdoor play behaviour using the Tool for Observing Play Outdoors (TOPO) developed by Loebach and Cox [49]. Environmental characteristics of the outdoor play space were captured from each centre's base map and additional variables, including gender and loose parts interaction, were integrated into the data collected within the OBM protocol. The reliability of the OBM method was measured by the degree of interrater reliability and agreement, using weighted κ and intraclass correlation coefficients [50,51]. All OBM points were collected at the centre-level, where children's play was assessed in relation to their outdoor space. This is in contrast to data collected at the child-level, where play behaviour is assessed in relation to individual children. OBM was implemented through centre-level data collection approaches to facilitate understanding of children's outdoor play at each participating ELCC centre. Twelve members of the research team were involved in collecting OBM data and participated in training sessions on the methodology prior to field work. A κ value of 0.918 (agreement = 95.9%) was achieved prior to beginning data collection. In addition, a 10% sample of data at each time point was double coded from recorded videos to further ensure interrater reliability and agreement.

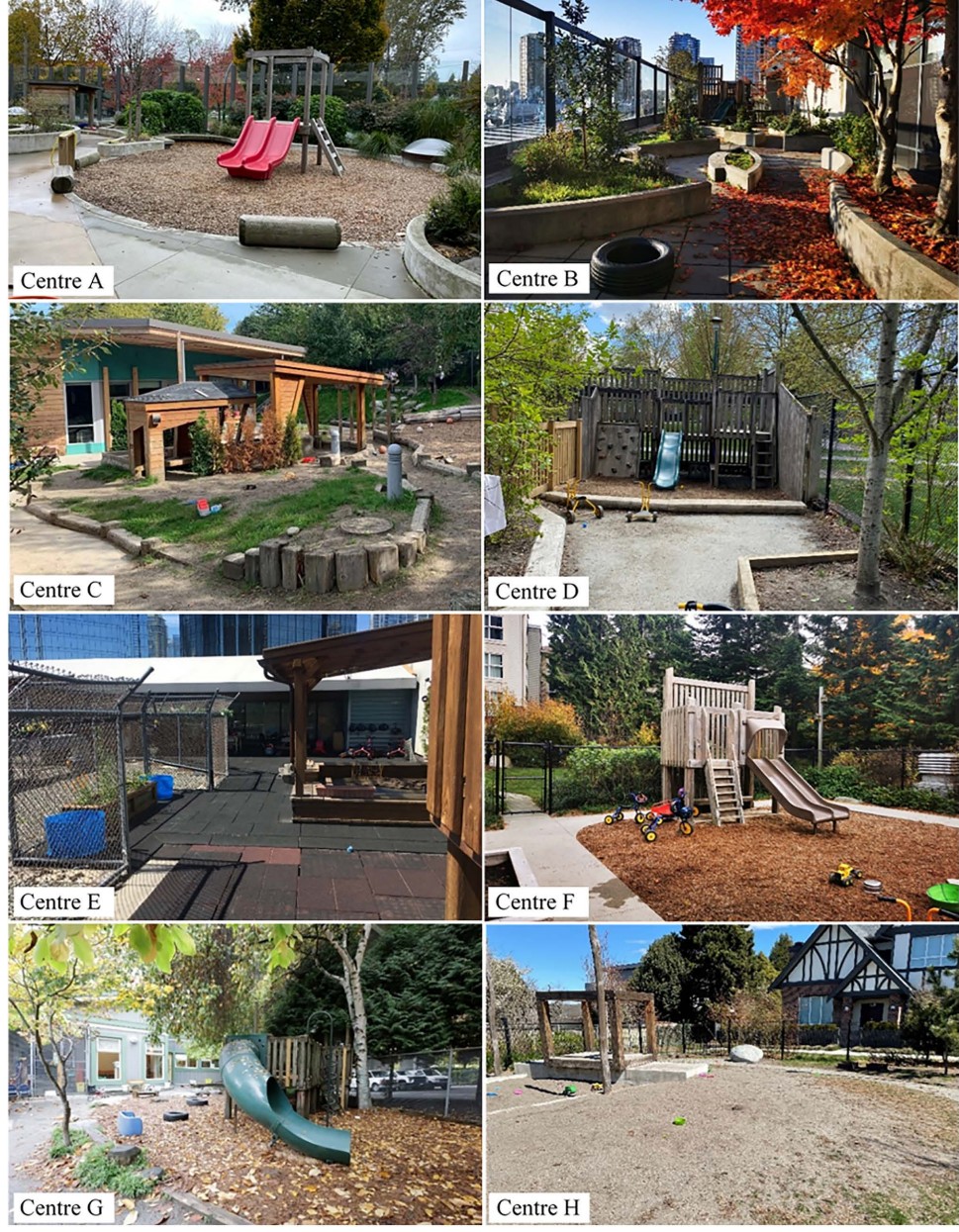

**Fig 1. Descriptive photos of participating ELCC centres (n = 8) at Time 1 [52].**

## Outcome variable

Children's outdoor play behaviour was coded using the expanded version of the TOPO [49]. The TOPO measures children's play behaviour through validated categories of 8 play types and 1 non-play type, along with their corresponding subtypes (Table 2). For this study, we did not code any digital play behaviour, and therefore, this TOPO play type category was not present within our dataset. A new dichotomous variable, *play participation,* was derived from the TOPO play types to understand differences in *play* vs *non-play* behaviour. The TOPO categories of physical play, exploratory play,

**Table 2. Tool for Observing Play Outdoors (TOPO) developed by Loebach and Cox [49].**

| Play Type | Description |
|---|---|
| Physical Play | Includes gross motor, fine motor, vestibular and rough and tumble. |
| Exploratory Play | Includes sensory, active and constructive. |
| Imaginative Play | Includes symbolic, sociodramatic and fantasy. |
| Play with Rules | Includes organic and conventional. |
| Bio Play | Includes plants, wildlife and care. |
| Expressive Play | Includes performance, artistic, language and conversation. |
| Restorative Play | Includes resting, retreat, reading and onlooking. |
| Non-play | Includes self-care, nutrition, distress, aggression, transition and other. |

imaginative play, play with rules, bio play and expressive play were categorized as *play*. The TOPO categories of restorative play and non-play were combined to create the *non-play* category because restorative play was often paired with non-play activities, specifically resting and onlooking. Up to three equally weighted TOPO codes could be assigned to each observational behaviour point, creating occurrences where an observation behaviour point could be considered *play* and *non-play* within the new *play participation* variable. Therefore, additional rules were determined to categorize observational behaviour points that were coded as non-play or restorative play and another play type (see S1 Appendix). The final primary outcome variable, *play participation*, categorized each observational behaviour point as either *play* (play occurred) or *non-play* (play did not occur). Further details of this process have been previously reported [52]. Each TOPO play type was also considered as an outcome variable in the secondary analysis.

### Primary explanatory variable

Categories for the primary explanatory variable, environmental play feature, were derived from previous research [22,28,53] and adapted based on shared commonalities across the eight participating ELCC centres in this study. The final environmental play feature variable included the following categories: sandbox, tricycle path, gardening area, outdoor stage, fixed equipment- playhouse, fixed equipment- climbing structure, fixed-equipment- water feature, and open area. All examined environmental play features were manufactured, with gardening areas containing natural materials but primarily fabricated with hard materials to enclose the plantings. These categories showcase that participating centres had limited natural play elements present, such as large trees or boulders. Portable equipment were captured within the loose parts variable also collected through the OBM protocol. Base maps of each participating ELCC centre were created to provide an overview of the environment at each participating centre and included information on topography, ground surface and environmental play features. Fig 2 displays an example of Centre A's base map with assigned environmental play feature areas while Fig 3 showcases Centre A's base map with observational behaviour points for Time 1 overlayed.

### Potential confounders and risk factors

Sociodemographic factors and play factors with known associations with children's play behaviour were also collected as part of the OBM protocol. Existing research identifies a relationship between children's outdoor play behaviour and gender [54–56], temperature and weather conditions [54,57,58], ground topography [59,60] and loose parts use [59,61]. Gender was collected during observations and recorded based on how the child presented using potential visible gender markers, as outlined elsewhere by Loebach et al. [62]. Data on temperature and weather conditions were recorded (www. timeanddate.com/weather) and matched to the day and time of data collection. Temperature was included as a continuous variable (ºC). Weather conditions were categorized into higher-level categories (cloudy, no rain; raining; sunny). Ground

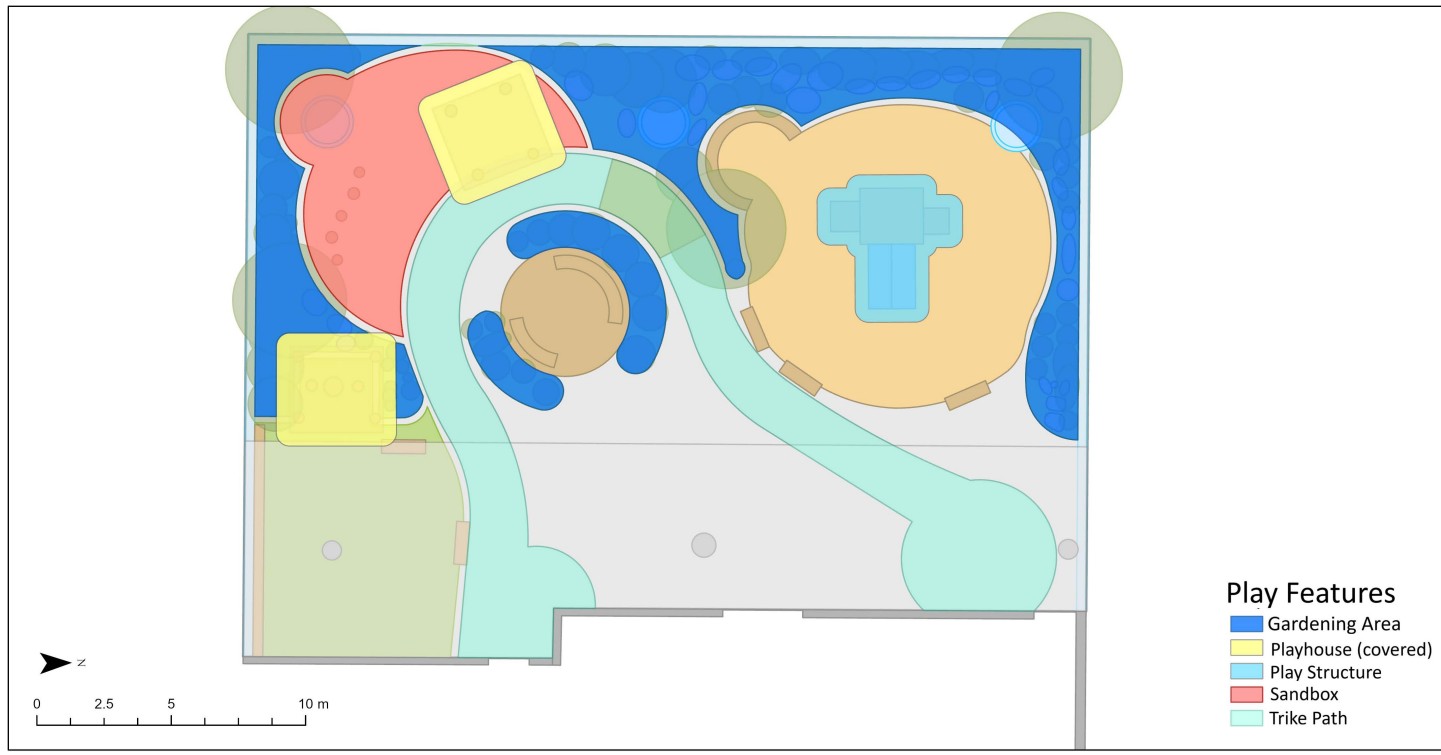

**Fig 2. Centre A base map with assigned outdoor play feature areas (non-highlighted areas are open areas).**

topography (no/very low slope; moderate slope; steep slope; uneven surface) was captured through each ELCC centre's base map. Interaction with loose parts (yes; no) was captured through the OBM protocol and considered children's interaction with natural and manufactured loose parts.

## Analysis

Analyses for this study sought to understand environmental play features associated with: 1) play participation (vs. non-play participation); and 2) the different play behaviour types captured through the TOPO (Table 2). Bivariate analyses were conducted to assess crude associations between play participation and environmental play features, and all covariates. The primary analysis used a multivariate logistic regression model to estimate the association between each environmental play feature and the dichotomous outcome variable, *play participation*, via odds ratio (OR) and 95% confidence intervals (CI). The influence of data collection time point on play behaviour was considered as a discreet, categorical variable (Time 1, Time 2, Time 3), however, the inclusion of time showed no significant difference in model fit and was therefore not included in the final model.

A secondary analysis considered each TOPO play category as a binary outcome variable (physical play, exploratory play, imaginative play, bio play, play with rules and expressive play). This investigation aimed to understand how environmental play features supported multiple forms of play, with the assumption that when children are afforded more opportunities for diverse forms of play to transpire, the play value of an environment is enhanced [38,39]. The TOPO categories of non-play and restorative play were not included in this analysis as they did not align with this study's definition of outdoor play. Bivariate analyses were completed to compare crude associations between play behaviour type and environmental play features. Each of the TOPO play types was looked at independently and the same regression model used for the

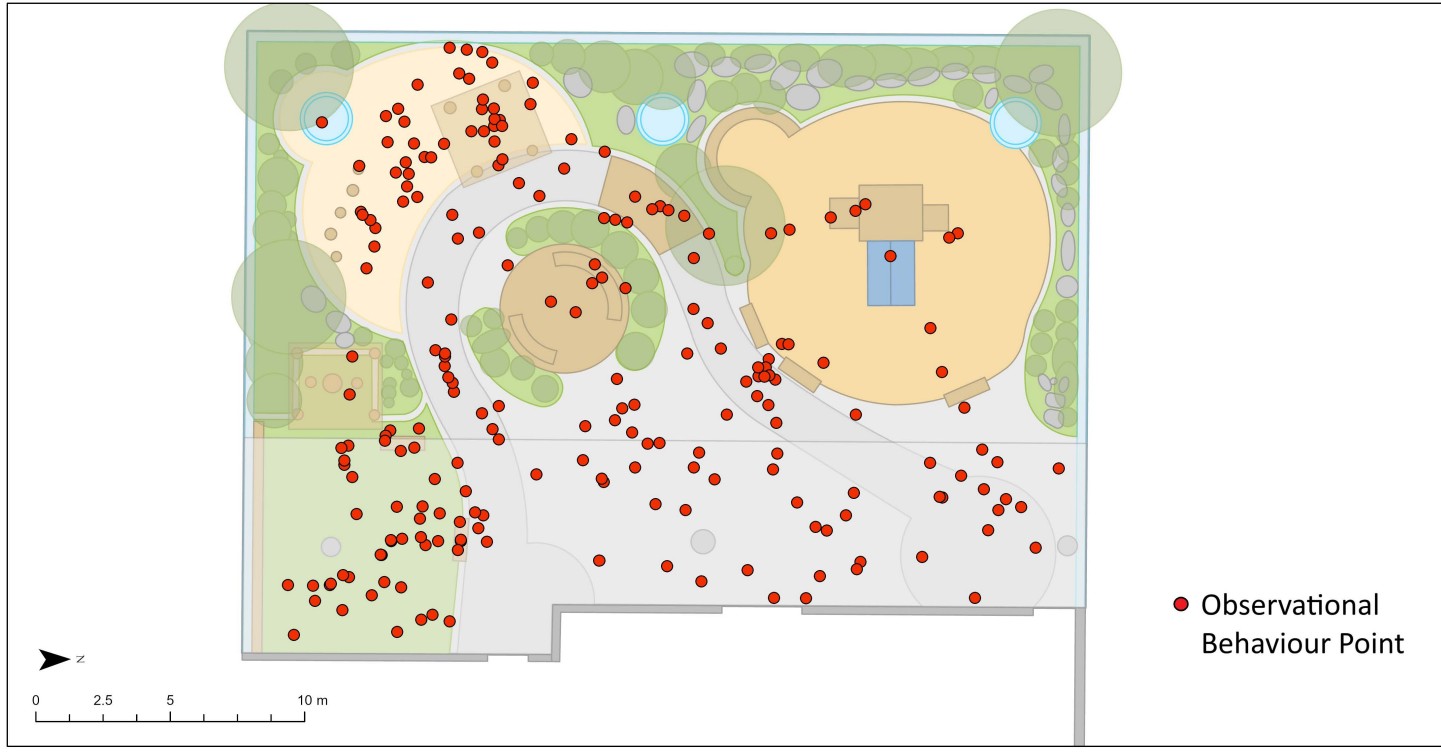

**Fig 3. Centre A base map with all observational behaviour points overlayed (Time 1).**

primary analysis was run for each play type as a binary outcome variable. The sample for this analysis was comprised of all play observations, where non-play participation observations were removed from the sample.

For both analyses, known confounders and risk factors identified a priori within the literature were included within the model. Multicollinearity was assessed using Variance Inflation Factors (VIF). Statistical analyses were performed using R-4.2.2.

## Results

### Study observations

A total of 5,213 observational behaviour points were collected across the eight participating ELCC centres and all time-points. Outside of open areas, where observations most frequently occurred (49.1%), tricycle paths (17.6%), sandboxes (12.0%) and climbing structures (9.4%) were most regularly used by children across all observational behaviour points. Children were observed most often in low topography terrain (89.9%), while steep (1.2%) and uneven terrains (8.9%) were less commonly observed across observations. This can be attributed to the low prevalence of steep and uneven terrain present across the eight participating ELCC centres. The majority of recorded observational behaviour points included the use of loose parts (75.3%).

Across all observational behaviour points (n = 5,213), children were observed participating in play 80.7% of the time (n = 4,206). Children's participation in play was most likely to occur in open areas (46.3%), tricycle paths (17.3%), sandboxes (13.3%) and fixed climbing structures (10.2%). Non-play observations were most common in open areas (61.0%) and on tricycle paths (18.8%). Table 3 presents the overall frequency distribution for our dataset, stratified by play and non-play participation type.

**Table 3. Descriptive sample characteristics of all observational behaviour points (n = 5,213), stratified by play participation type (play or non-play).**

| Variable | Stratified by Play Participation Type | | | All Observational Behaviour Points |
|---|---|---|---|---|
| | **Non-Play** | **Play** | **p-value** | |
| Number of Play Observations | 1,007 | 4,206 | | 5,213 |
| Environmental Play Feature [n (%)] | | | <0.001 | |
| Open Area | 614 (61.0) | 1,948 (46.3) | | 2,562 (49.1) |
| Gardening Area | 26 (2.6) | 219 (5.2) | | 245 (4.7) |
| Fixed Equipment - Playhouse | 39 (3.9) | 240 (5.7) | | 279 (5.4) |
| Fixed Equipment - Climbing Structure | 61 (6.1) | 431 (10.2) | | 492 (9.4) |
| Sandbox | 69 (6.9) | 558 (13.3) | | 627 (12.0) |
| Outdoor Stage | 7 (0.7) | 49 (1.2) | | 56 (1.1) |
| Tricycle Path | 189 (18.8) | 727 (17.3) | | 916 (17.6) |
| Fixed Equipment - Water Feature | 2 (0.2) | 34 (0.8) | | 36 (0.7) |
| Topography [n (%)] | | | <0.001 | |
| Low | 938 (93.1) | 3,750 (89.2) | | 4,688 (89.9) |
| Steep | 4 (0.4) | 59 (1.4) | | 63 (1.2) |
| Uneven | 65 (6.5) | 397 (9.4) | | 462 (8.9) |
| Weather Conditions [n (%)] | | | 0.001 | |
| Sunny | 202 (20.1) | 944 (22.4) | | 1,146 (22.0) |
| Cloudy, no rain | 607 (60.3) | 2625 (62.4) | | 3,232 (62.0) |
| Raining | 198 (19.7) | 637 (15.1) | | 835 (16.0) |
| Temperature (Mean (SD)) | 8.57 (3.33) | 9.10 (3.13) | <0.001 | 9.00 (3.18) |
| Gender (girl) [n (%)] | 444 (44.1) | 1,715 (40.8) | 0.06 | 2,159 (41.4) |
| Loose Parts Use (yes) [n (%)] | 588 (58.4) | 3,337 (79.3) | <0.001 | 3,925 (75.3) |

Within the play observations (n = 4,206), the most frequent play types children participated in were physical play (76.0%) and exploratory play (44.0%). While open areas were the most frequently used spaces among all types of play, a large portion of bio play observations occurred in gardening areas (29.5%), including raised planters and at-grade gardening areas. In addition, bio play observations were observed often on uneven terrain (25.9%), which contrasts the low frequency of all observational behaviour points (8.9%) occurring on uneven terrain. While most play types involved the use of loose parts, play with rules did not have a large proportion of loose parts use (44.0%), in comparison to all observational behaviour points (75.3%). Exploratory play (94.6%) and bio play (87.1%) had the highest frequency of loose parts use among all play types. Table 4 provides an overview of the frequency of environmental play feature usage and associated covariates, by play behaviour type.

## Association between play participation and environmental play features

The results of the multivariate logistic regression model examining the association between play participation and environmental play features is presented in Table 5. In comparison to open areas, all investigated environmental play features, except for outdoor stages, were significantly associated with children's play participation. When adjusting for topography, weather conditions, temperature, gender and loose parts interaction, fixed water features (OR:6.30, 95% CI = 1.48, 26.76), play structures (OR:3.01, 95% CI = 2.24, 4.04) and gardening areas (OR:2.68, 95% CI = 1.75, 4.10) had the highest odds of play participation. Steep slopes (OR:4.17, 95% CI = 1.48, 11.75), uneven terrain (OR:1.61, 95% CI = 1.21, 2.15), temperature (OR:1.04, 95% CI = 1.02, 1.06) and loose parts interaction (OR:3.01, 95% CI = 2.58, 3.51) were significantly associated with children's play participation within the adjusted model. In addition, raining weather conditions were significantly

**Table 4. Descriptive sample characteristics of all play observations (n = 4,206), stratified by play behaviour type.184.**

| Variable | Play Behaviour Type | | | | | |
|---|---|---|---|---|---|---|
| | Physical Play | Exploratory Play | Imaginative Play | Play with Rules | Bio Play | Expressive Play |
| Number of Play Observations | 3,196 | 1,849 | 405 | 184 | 224 | 891 |
| Environmental Play Feature [n(%)] | | | | | | |
| Open Area | 1,472 (46.1) | 772 (41.8) | 178 (44.0) | 96 (52.2) | 94 (42.0) | 494 (55.4) |
| Gardening Area | 133 (4.2) | 92 (5.0) | 20 (5.0) | 8 (4.3) | 66 (29.5) | 37 (4.2) |
| Fixed Equipment - Playhouse | 175 (5.5) | 136 (7.4) | 30 (7.4) | 1 (0.5) | 0 (0.0) | 55 (6.2) |
| Fixed Equipment - Climbing Structure | 352 (11.1) | 109 (5.9) | 62 (15.3) | 20 (10.9) | 5 (2.2) | 83 (9.3) |
| Sandbox | 413 (12.9) | 422 (22.8) | 51 (12.6) | 5 (2.7) | 10 (4.5) | 86 (9.7) |
| Outdoor Stage | 39 (1.2) | 33 (1.8) | 7 (1.7) | 0 (0.0) | 0 (0.0) | 9 (1.0) |
| Tricycle Path | 590 (18.5) | 258 (14.0) | 56 (13.9) | 54 (29.3) | 48 (21.4) | 124 (13.9) |
| Fixed Equipment - Water Feature | 22 (0.7) | 27 (1.5) | 1 (0.2) | 0 (0.0) | 1 (0.4) | 3 (0.3) |
| Topography [n(%)] | | | | | | |
| Low | 2,868 (89.7) | 1,651 (89.3) | 367 (90.6) | 170 (92.4) | 158 (70.5) | 816 (91.6) |
| Steep | 47 (1.5) | 12 (0.6) | 6 (1.5) | 4 (2.2) | 8 (3.6) | 9 (1.0) |
| Uneven | 281 (8.8) | 186 (10.1) | 32 (7.9) | 10 (5.4) | 58 (25.9) | 66 (7.4) |
| Weather Conditions [n(%)] | | | | | | |
| Sunny | 726 (22.7) | 397 (21.5) | 90 (22.2) | 48 (26.1) | 53 (23.7) | 191 (21.4) |
| Cloudy, no rain | 2,000 (62.6) | 1,107 (59.9) | 258 (63.7) | 121 (65.8) | 149 (66.5) | 573 (64.3) |
| Raining | 470 (14.7) | 345 (18.7) | 57 (14.1) | 15 (8.2) | 22 (9.8) | 127 (14.3) |
| Temperature [mean(SD)] | 9.02 (3.16) | 9.32 (3.12) | 9.13 (3.06) | 8.82 (3.23) | 10.03 (2.74) | 8.91 (3.11) |
| Gender (girl) [n(%)] | 1,273 (39.8) | 715 (38.7) | 179 (44.2) | 72 (39.1) | 99 (44.2) | 399 (44.8) |
| Loose Parts Use (yes) [n(%)] | 2,540 (79.5) | 1,750 (94.6) | 312 (77.0) | 81 (44.0) | 195 (87.1) | 638 (71.6) |

[a]Play observations can be coded as (up to) 3 play type categories and therefore, the sum of observations across play types exceeds the sample size (n = 4,206).

associated with decreased likelihood of play participation (OR: 0.73, 95% CI = 0.58, 0.92). These findings indicate that environmental play features beyond outdoor open areas provide significant play worth to children's outdoor play. Further analysis of the type of play afforded by each environmental play feature is warranted.

## Associations between play behaviour type and environmental play features

Table 6 outlines the results of the adjusted multivariate logistic regression model examining each play behaviour type and environmental play features. Unadjusted results for each play type outcome are available in S2 Appendix. Fixed climbing structures (OR:2.23, 95% CI = 1.79, 2.78), sandboxes (OR:1.29, 95% CI = 1.07, 1.56) and tricycle paths (OR:1.33, 95% CI = 1.13, 1.56) were positively associated with children's physical play at ELCC centres in comparison to open areas. Rainy weather conditions (OR:0.76, 95% CI = 0.63, 0.92), steep slope terrain (OR:2.01, 95% CI = 1.10, 3.65) and the use of loose parts (OR:1.92, 95% CI = 1.68, 2.19) were all significantly associated with children's participation in physical play. Girls were less likely to participate in physical play then boys (OR:0.84, 95% CI = 0.75, 0.94).

Gardening areas (OR:1.51, 95% CI = 1.12, 2.03), fixed playhouses (OR: 2.09, 95% CI = 1.60, 2.73), sandboxes (OR:4.52, 95% CI = 3.70, 5.53), outdoor stages (OR:2.56, 95% CI = 1.46, 4.52) and fixed water features (OR:11.74, 95% CI = 4.86, 28.33) were significantly associated with children's exploratory play at ELCC in comparison to open areas. Uneven terrain (OR:1.60, 95% CI = 1.28, 2.00), rainy weather conditions (OR:1.46, 95% CI = 1.19, 1.80), temperature (OR:1.04, 95% CI = 1.02, 1.06) and loose part use (OR:9.51, 95% CI = 7.60, 11.89) were also significantly associated with

**Table 5. Univariate and multivariate logistic regression results (OR, 95% CI) examining the association between play participation and environmental play feature.**

| Variable | Play Participation | |
|---|---|---|
| | Unadj. OR[a] [95% CI] | Adj. OR [95% CI][b] |
| Environmental Play Feature | | |
| Open Area | Ref | Ref |
| Gardening Area | 2.65 [1.75;4.03]*** | 2.68 [1.75;4.10]*** |
| Fixed Equipment - Playhouse | 1.94 [1.37;2.75]*** | 1.86 [1.30;2.67]*** |
| Fixed Equipment - Climbing Structure | 2.23 [1.68;2.96]*** | 3.01 [2.24;4.04]*** |
| Sandbox | 2.55 [1.95;3.33]*** | 2.19 [1.67;2.87]*** |
| Outdoor Stage | 2.21 [0.99;4.90] | 1.55 [0.69;3.48] |
| Tricycle Path | 1.21 [1.01;1.46]* | 1.23 [1.02;1.49]* |
| Fixed Equipment - Water Feature | 5.36 [1.28;22.37]* | 6.30 [1.48;26.76]* |
| Topography | | |
| Low/ No Slope | Ref | Ref |
| Steep Slope | 3.69 [1.34;10.18]* | 4.17 [1.48;11.75]** |
| Uneven Terrain | 1.53 [1.16;2.00]** | 1.61 [1.21;2.15]** |
| Weather Conditions | | |
| Sunny | Ref | Ref |
| Cloudy, no rain | 0.93 [0.78;1.10] | 0.90 [0.75;1.08] |
| Raining | 0.69 [0.55;0.86]*** | 0.73 [0.58;0.92]** |
| Temperature | 1.05 [1.03;1.07]* | 1.04 [1.02;1.06]*** |
| Gender | | |
| Boy | Ref | Ref |
| Girl | 0.87 [0.76;1.00] | 0.90 [0.78;1.04] |
| Loose Part Interaction | | |
| No | Ref | Ref |
| Yes | 2.74 [2.36;3.17]*** | 3.01 [2.58;3.51]*** |

Adj. = adjusted; Unadj. = unadjusted; OR = odds ratio; 95% CI = 95% confidence interval.

Note: significance level, *: p < 0.05, **: p < 0.01, ***: p < 0.001.

[a]Unadjusted OR's calculated via 6 bivariate logistic regression models examining play participation and each covariate independently.

[b]OR's from multivariate logistic regression, adjusting for topography, weather conditions, temperature, gender and loose part interaction.

exploratory play participation within the adjusted model. Girls were less likely to participate in exploratory play then boys (OR:0.86, 95% CI = 0.76, 0.98).

Fixed playhouses (OR: 1.57, 95% CI = 1.04, 2.37) and climbing structures (OR:2.00, 95% CI = 1.46, 2.74) were positively associated with children's imaginative play at ELCC in comparison to open areas. Gender, topography, weather conditions, temperature and loose parts interaction were not significantly associated with children's participation in imaginative play. Tricycle paths (OR:1.61, 95% CI = 1.14, 2.29) were significantly associated with increasing children's play with rules in comparison to open areas. In contrast, playhouses (OR:0.11, 95% CI = 0.01, 0.76) and sandboxes (OR:0.25, 95% CI = 0.10, 0.63) were negatively associated with children's participation in play with rules activities. Due to low case numbers, associations between play with rules and outdoor stages and fixed water features was not possible to calculate. Children's participation in play with rules was negatively associated with uneven terrain (OR:0.51, 95% CI = 0.26,0.99),

**Table 6. Multivariate logistic regression results (OR, 95% CI) examining the association between play behaviour types and environmental play feature.**

| Variable | Physical Play Adj. OR [95% CI][a] | Exploratory Play Adj. OR [95% CI][a] | Imaginative Play Adj. OR [95% CI][a] | Play with Rules Adj. OR [95% CI][a] | Bio Play Adj. OR [95% CI][a] | Expressive Play Adj. OR [95% CI][a] |
|---|---|---|---|---|---|---|
| Environmental Play Feature | | | | | | |
| Open Area | Ref | Ref | Ref | Ref | Ref | Ref |
| Gardening Area | 0.87 [0.67;1.14] | 1.51 [1.12;2.03]** | 1.22 [0.75;1.98] | 0.88 [0.42;1.86] | 9.38 [6.46;13.63]*** | 0.75 [0.52;1.08] |
| Fixed Equipment - Playhouse | 1.18 [0.91;1.53] | 2.09 [1.60;2.73]*** | 1.57 [1.04;2.37]* | 0.11 [0.01;0.76]* | - | 1.03 [0.75;1.41] |
| Fixed Equipment - Climbing Structure | 2.23 [1.79;2.78]*** | 1.08 [0.84;1.39] | 2.00 [1.46;2.74]*** | 0.68 [0.41;1.13] | 0.28 [0.11;0.71]** | 0.79 [0.61;1.03] |
| Sandbox | 1.29 [1.07;1.56]** | 4.52 [3.70;5.53]*** | 1.14 [0.82;1.58] | 0.25 [0.10;0.63]** | 0.41 [0.21;0.80]** | 0.67 [0.53;0.87]** |
| Outdoor Stage | 1.47 [0.82;2.63] | 2.56 [1.46;4.52]** | 1.90 [0.84;4.30] | - | - | 0.86 [0.42;1.78] |
| Tricycle Path | 1.33 [1.13;1.56]*** | 0.89 [0.75;1.06] | 0.86 [0.63;1.18] | 1.61 [1.14;2.29]** | 1.63 [1.13;2.36]** | 0.65 [0.52;0.80]*** |
| Fixed Equipment - Water Feature | 1.24 [0.63;2.46] | 11.74 [4.86;28.33]*** | 0.40 [0.05;2.93] | - | 0.62 [0.08;4.76] | 0.39 [0.12;1.29] |
| Topography | | | | | | |
| Low/ No Slope | Ref | Ref | Ref | Ref | Ref | Ref |
| Steep Slope | 2.01 [1.10;3.65]* | 0.67 [0.34;1.30] | 1.03 [0.43;2.43] | 1.52 [0.53;4.35] | 5.81 [2.56;13.21]*** | 0.70 [0.34;1.43] |
| Uneven Terrain | 1.09 [0.89;1.34] | 1.60 [1.28;2.00]*** | 0.84 [0.57;1.24] | 0.51 [0.26;0.99]* | 3.30 [2.32;4.69]*** | 0.76 [0.57;1.00]* |
| Weather Conditions | | | | | | |
| Sunny | Ref | Ref | Ref | Ref | Ref | Ref |
| Cloudy, no rain | 0.95 [0.82;1.09] | 0.94 [0.81;1.10] | 1.00 [0.78;1.29] | 0.87 [0.62;1.24] | 0.86 [0.61;1.21] | 1.08 [0.90;1.30] |
| Raining | 0.76 [0.63;0.92]** | 1.46 [1.19;1.80]*** | 0.90 [0.63;1.27] | 0.40 [0.22;0.72]** | 0.68 [0.40;1.14] | 0.86 [0.68;1.11] |
| Temperature | 1.00 [0.98;1.02] | 1.04 [1.02;1.06]*** | 1.01 [0.98;1.05] | 0.99 [0.95;1.04] | 1.13 [1.07;1.19]*** | 0.99 [0.97;1.02] |
| Gender | | | | | | |
| Boy | Ref | Ref | Ref | Ref | Ref | Ref |
| Girl | 0.84 [0.75;0.94]** | 0.86 [0.76;0.98]* | 1.12 [0.91;1.38] | 0.85 [0.63;1.16] | 1.17 [0.88;1.55] | 1.17 [1.01;1.35]* |
| Loose Part Interaction | | | | | | |
| No | Ref | Ref | Ref | Ref | Ref | Ref |
| Yes | 1.92 [1.68;2.19]*** | 9.51 [7.60;11.89]*** | 1.20 [0.93;1.55] | 0.25 [0.18;0.34]*** | 2.75 [1.81;4.19]*** | 0.79 [0.67;0.94]** |

Adj. = adjusted; Unadj. = unadjusted; OR = odds ratio; 95% CI = 95% confidence interval

Note: significance level, *: p < 0.05, **: p < 0.01, ***: p < 0.001

[a]OR's from multivariate logistic regression, adjusting for topography, weather conditions, temperature, gender and loose part interaction.

rainy weather conditions (OR:0.40, 95% CI = 0.22, 0.72) and the use of loose parts (0.25, 95% CI = 0.18, 0.34). Gender was not significantly associated with participation in play with rules.

Gardening areas (OR:9.38, 95% CI = 6.46, 13.63) and tricycle paths (OR:1.63, 95% CI = 1.13,2.36) were significantly positively associated with children's bio play in comparison to open areas. Climbing structures (OR:0.28, 95% CI = 0.11, 0.71) and sandboxes (OR:0.41, 95% CI = 0.21, 0.80) were negatively associated with bio play opportunities. In comparison to low slope terrain, steep slopes (OR:5.81, 95% CI = 2.56, 13.21) and uneven terrains (OR:3.30, 95% CI = 2.32, 4.69) were significantly associated with increased bio play participation. In addition, interactions with loose parts increased the odds of children's participation in bio play (OR:2.75, 95% CI = 1.81, 4.19). Diverse weather conditions were not associated with bio play; however, bio play did increase in higher temperatures (OR:1.13, 95% CI = 1.07, 1.19). There were no significant gender differences in bio play participation.

In comparison to open areas, sandboxes (OR:0.67, 95% CI = 0.53, 0.87) and tricycle paths (OR:0.65, 95% CI = 0.52, 0.80) were negatively associated with children's participation in expressive play. There were no significant positive associations with expressive play and environmental play features, indicating that open areas may provide benefits for expressive play participation. Uneven terrain (OR:0.76, 95% CI = 0.57, 1.00) and the use of loose parts (OR:0.79, 95% CI = 0.67, 0.94) were negatively associated with participation in expressive play. Girls were significantly more likely to participate in expressive play than boys (OR:1.17, 95% CI = 1.01, 1.35).

## Discussion

This study aimed to understand associations between children's play behaviour and common environmental play features located across eight urban ELCC centres in the Greater Vancouver region. Descriptive findings underline open areas, sandboxes, climbing structures and tricycle paths as the most frequented spaces where outdoor play transpired. This consideration likely reflects the prominent size of these features in the overall space, their consistent presence across the participating ELCC centres, and children's familiarity with them. Comparable patterns have been observed in children's exploratory behaviours within outdoor play environments at ELCC centres [24], highlighting the role of prior experiences and object familiarity in shaping perceived affordances [41]. Aside from outdoor stages, all examined play features offered significantly more play opportunities for children in comparison to open spaces. The results of this study align with previous observational research examining children's use of outdoor play features and microspaces [37,63,64]. Manufactured fixed equipment, including climbing structures and playhouses, are commonly identified as children's preferred play locations [63,64] and observational studies have highlighted preschool-aged children's affinity for riding toy pathways and sandboxes [64].

### Evaluating the value of play features through the theory of affordances

Children's actualized affordances for outdoor play are environmental opportunities that are both perceived and acted upon by the child, resulting in meaningful and varied play behaviours [42]. The presence of affordances that offer diverse play opportunities reflects the environment's capacity to support a wide range of developmental needs and interests [45]. The findings of the secondary analysis, summarized in Table 7, highlight associations between each TOPO play behaviour type and examined environmental play feature, offering valuable insights into how each play feature may afford multiple forms of play to transpire. Gardening areas, playhouses, climbing structures and tricycle paths were environmental play features that offered a positive association with multiple play behaviour types, indicating that these features provided affordances for diverse forms of play. Sandboxes offered limited diverse play experiences in comparison to open areas. The play value of each examined environmental play features, defined by the diversity of play afforded, are examined further below, providing important knowledge to inform future ELCC outdoor play space design and planning.

Table 7. Summary of outdoor play type and environmental play feature associations.

| Play Behaviour Type | Environmental Play Feature Associations (in comparison to open areas) | | | | | | |
|---|---|---|---|---|---|---|---|
| | Gardening Area | Playhouse | Climbing Structure | Sandbox | Outdoor Stage | Tricycle Path | Water Feature |
| Physical | | | + | + | | + | |
| Exploratory | + | + | | + | + | | + |
| Imaginative | | + | + | | | | |
| Play with Rules | | − | | − | | + | |
| Bio Play | + | | − | − | | + | |
| Expressive Play | | | | − | | − | |

### Tricycle paths

Tricycle paths were associated with multiple play types, providing opportunities for children to chase one another, partake in games and gather natural elements found in intersecting garden areas. Tricycle paths frequently represented the sole opportunities for children to engage in high-speed play, such as running or using wheeled equipment, a finding echoed in existing literature linking tricycle paths with elevated levels of physical activity in children [28,65,66]. The positive association between participation in rules-based games and tricycle paths can be attributed to the smooth, flat surfaces these paths provide for mobile equipment and the perceived safety of these areas to partake in racing, tag or other structured games [67]. Children's engagement in bio play in association with tricycle pathways observed in this study is less well-defined; the presence of garden areas in close proximity to tricycle paths at participating ELCC centres may have facilitated children's collection, transport and use of natural materials along and into these pathways during play experiences. This concept of cross-area play is supported by findings from Kuh [68], where children were observed carrying natural materials, such as pinecones and pine needles, into fabricated play areas, including play houses and tricycle paths.

### Climbing structures

All participating ELCC centres in this study had a climbing structure of some form. These structures were positively associated with children's physical and imaginative play opportunities, providing affordances for climbing, balancing and other gross motor activities alongside make-belief and creative play. Previous research [20,37,63] supports the importance of fixed play structures on children's functional play, including running, jumping, spinning, balancing and climbing. In addition, the presence of fixed play equipment and playground structures support children's physical activity and movement skills [26,28,65,66]. The use of climbing structures to support imaginative play is most likely when the structure contains a theme, such as a castle or a ship [20,28,37]. Although the climbing structures in this study were not thematically designed, their open-ended qualities encouraged children to construct their own imaginative interpretations, consistent with findings from other research [69–71]. Previous research has highlighted the significance of play structures as spaces that afford opportunities for refuge or withdrawal from the surrounding activity and social dynamics [72,73]. The absence of significant associations with play types aside from physical and imaginative play suggest that children's concealment or retreat were not commonly enacted in relation to these climbing structures in this study.

### Sandboxes

Sandboxes were among the most utilized environmental play features in our study, however, they provided limited affordances for diverse play opportunities. The sandbox is often designed as a 'catch-all' spot where all children can partake in a preferred activity and it is designed to facilitate individual and group play [74]. Our study showcased that the sandbox was positively associated with exploratory and physical play opportunities, likely due to the use of loose parts in conjunction with the sandbox. Constructive and manipulative play have been previously linked to sand play settings [18,35]. Opportunities for physical play were primarily from fine motor activities navigating and manipulating the loose parts in the sandbox. However, other forms of play were limited in sandbox play features, leading us to assume the sandbox may not resonate with all children, or children may use this play feature in an inconsistent way. Likewise, Jarrett et al. [74] observed that engagement with the sandbox tends to be polarized, with children either using it frequently or not at all. Previous research suggests that the capacity of sandboxes to support diverse forms of play depends on their thoughtful integration with other natural elements and loose parts [74]. To effectively foster a broader range of play behaviours, these spaces may require varied types of loose and natural materials then what was observed in this study.

### Playhouses

Playhouses were positively associated with exploratory and imaginative play, providing opportunities for children to participate in sociodramatic activities, often involving loose parts, such as pretending to bake muffins in a pretend kitchen.

These fixed structures offer blank canvases for children's pretend play, supporting a range of social role experimentation and creative interactions with natural elements and peers [24,75]. These findings replicate previous studies demonstrating the significance of enclosed spaces on children's dramatic play [20,28,37]. Most participating ELCC centres in this study included playhouses situated adjacent to sandboxes and provided opportunities for explorative play, such as sand and water activities, in a sheltered location. It's evident that playhouses demonstrate significant potential to stimulate children's creativity, however, this study highlights that their use is limited in supporting physical activity, peer-interactive games, or engagement with natural elements. Enclosed spaces are essential in outdoor play spaces, offering opportunities for hiding during peer games, providing potential refuge, or simply affording a reprieve from the density of activity in the area [73]. However, the confined nature of these structures may limit their support for physical activities or games that necessitate movement, as such activities typically require larger, open spaces where children can visibly see the space beyond [76].

### Water features

Among sampled ELCC centres, fixed water features, including water pumps, hoses and water troughs, were often installed to support gardening activities and showcase environmental solutions to sustainability. However, water features were not significantly associated with most forms of play in this study, including bio play opportunities, and children were rarely observed interacting with these play features across all observations. This supports the notion that children were not engaging in planting or gardening activities with the use of these water features. Through our observations, we noted that all water features were hard to use without the support of an educator. Children often struggled to navigate the water pumps on their own and required assistance by an educator, or at times a peer, to initiate water play. It was also noted that water features were often closed for parts of the year due to weather conditions, such as freezing temperatures. Although water play features are recognized as essential components in ELCC outdoor space design [77], less is known on child-friendly design approaches to support their independent use. To facilitate bio play and more diverse play opportunities, fixed water features must be easily accessible to children without the support of an educator.

### Gardening areas

This study underscores the significance of gardening areas as one of the primary environmental features that stimulate substantial bio-play opportunities, positioning them as a crucial element in children's interactions with nature. This aligns with the intuitive understanding that gardening spaces are typically designed to promote children's engagement with natural materials [68]. Additionally, prior research has established that gardening areas are closely associated with a variety of play types, including creative play [78,79], physical play [80,81], and cooperative play among peers [82,83]. Gardening areas can evoke storytelling and role-playing in children, alongside the physical movement required for gardening-related tasks [79]. In this study, the limited use of gardening areas, along with their null association with a variety of play types, suggests that these play features may not effectively support engaging and dynamic play experiences for children. The gardening areas examined were predominantly composed of raised concrete planters, which may have encouraged more static and directive forms of play due to their closed-ended design. Additionally, the limited use of gardening areas by children could be attributed to restrictions on materials and the use of these spaces set in place by ELCC centres or educators, such as not stepping on plantings, as described in other studies [83]. When gardening areas are designed primarily for adult's horticultural purposes, opportunities for children's engagement and diverse play experiences are limited.

### Outdoor stages

The environmental play feature category, outdoor stage, was formed to highlight the performative spaces used for singing, dancing, acting or expressive circle times found within this study's participating centres, and within the larger ELCC landscape [84]. However, this study did not find associations between children's expressive play and raised stage areas. Children frequently used these areas as extensions of the open play space, bringing loose parts and portable equipment

onto the stage features, a practice that reflects their engagement in exploratory play. These spaces were also not clearly outlined as alluring performative and expressive play opportunities and required the educators to direct or initiate these play forms for them to take place. To support more affordances for diverse play to occur, further consideration of how outdoor stages are set-up and mobilized within ELCC programming is required. These results question if it is necessary to implement a dedicated space for a stage, or if this form of play can occur naturally within other play areas or by provocations in open areas.

### Open areas

Open areas represented the largest surface area and were the most frequented areas by children across the participating ELCC centres. The open-ended nature of these spaces contributes to flexible use, integration with portable materials and larger spaces for movement [22,77]. Compared to other environmental play features, this study found that open areas provided greater opportunities for expressive play, bio play and play with rules. These spaces were often intentionally designed to foster social interaction and creative expression, incorporating elements such as tables and portable art easels. Natural materials were also present, largely due to the proximity of adjacent gardening areas. However, the findings suggest that while open areas may support artistic activities and natural loose parts play, these play features may not offer the same breadth of physical play experiences as other environmental features examined in this study. Outdoor open areas in ELCC centres are routinely designed as circulation spaces for physical play (moving from space to space), and to encourage children's large body movements or rough and tumble play. Even with large amounts of space available in open areas, children in this study were more likely to participate in physical play opportunities at climbing structures and on tricycle paths. The open areas examined may have offered minimal affordances for physical play based on the way in which the space was set up for children to engage with. As these areas do not have fixed equipment or designated purposes, the available provocations set up in the spaces will guide the behaviours of children [85]. Refshauge et al. [37] found that children's participation in non-play activities frequently took place in open gathering spaces due to the configuration of seating elements, often encouraging eating activities to take place in these settings. While further research is required to understand the limitations of open outdoor areas at ELCC centres, the results of this study suggest that open spaces may not entice children to participate in physical play opportunities without the proper provocations in place.

The findings of this study offer important contributions to the understanding of how specific environmental features within outdoor play spaces can support diverse forms of children's play. Through an affordance theory lens, this study provides valuable insights into the play potential of elements commonly found in ELCC outdoor play spaces. Although the significance of high-quality outdoor environments on children's development is increasingly acknowledged, the deliberate selection and integration of play features that support a wide spectrum of play behaviours remain underemphasized. This gap persists despite longstanding evidence that physical characteristics of outdoor environments, such as spatial configuration and material variety, are key barriers to meeting quality outdoor play experiences in ELCC centres [86,87]. Given the complex and often competing demands of space allocation in urban ELCC design, it is critical to prioritize outdoor environments that maximize play affordances and inclusivity.

### Strengths and limitations

This study provides new knowledge on affordances for diverse outdoor play opportunities among children aged 2–6 in urban ELCC centres. While previous studies have explored physical outdoor spaces in relation to children's behaviours and movements, this study is, to our knowledge, the first to evaluate specific environmental play features using a comprehensive measurement approach to outdoor play. A strength of this study is the categorization of environmental play features and characteristics across eight participating ELCC centres to support an understanding of common features routinely present in urban ELCC centres. The derived play feature categories (sandbox, tricycle path, gardening area, outdoor stage, fixed equipment- playhouse, fixed equipment- climbing structure, fixed-equipment- water feature, and

open area) are also reflective of existing literature examining urban ELCC environments in other geographic contexts [24,63], supporting generalizability of this variable in future studies. While this study aimed to identify commonalities in play features across urban ELCCs, site-specific differences were present, such as topography, ground surfaces, and the level of challenge offered by climbing structures, and were not accounted for in this analysis. Further examination of each participating ELCC centre individually could offer a deeper understanding of how unique environmental conditions shape children's outdoor play.

This study mobilized novel observation methods to capture children's diverse play behaviour, creating opportunities to examine play beyond the traditional confines of physical activity measurement techniques. The use of behaviour mapping relies on researcher observations to record, track and code children's play behaviour, offering the potential for researcher subjectivity and discrepancies amongst coders. To account for this possible limitation, rigorous training and double coding procedures were employed to ensure reliability amongst the research team. Nonetheless, an element of subjectivity may exist and future analyses within the larger PRO-ECO study will examine children's qualitative perspectives of the examined outdoor play environments to provide a child-centered perspective. Additionally, the current study did not examine variations in play behaviour among children with diverse developmental abilities and needs. Future research would benefit from incorporating considerations of inclusivity in the design and evaluation of environmental play features.

## Conclusion

The results of this study provide important findings on the play value of outdoor affordances that are commonly found in urban ELCC centres in the Greater Vancouver region. Open areas are often the largest and most frequented spaces within the examined outdoor play settings, however, they provided minimal affordances for physical play in comparison to other measured outdoor play features. Consideration of how outdoor open areas are set up and supported by educators is important to enhance these spaces and support more diverse play opportunities. Opportunities for children's engagement with natural materials were limited and primarily occurred in gardening areas, or play features adjacent to gardening areas. Further integration of natural elements beyond the confines of gardening areas are essential to support increased engagement with natural materials.

Future research should consider the impact of movement across environment play features and areas, acknowledging that children may diversify their play behaviour by using multiple spaces and features of the environment. Furthermore, further research would benefit from considering the role of loose parts as a mediator in the play behaviour and environmental play feature relationship. Nonetheless, the results presented within this study provide critical guidance for outdoor space design within ELCC settings.

## Supporting information

**S1 Appendix. PRO-ECO play/non-play variable rules.**
(DOCX)

**S2 Appendix. Bivariate logistic regression results (OR, 95% CI) examining the association between each outcome and covariates.**
(DOCX)

## Acknowledgments

We extend our appreciation to the children who participated in this study and provided their valuable thoughts and ideas to contribute to this project. We thank the YMCA for their partnership on this study and the members of the Outside Play Lab that supported data collection throughout the PRO-ECO study to inform this manuscript.

# Author contributions

**Conceptualization:** Rachel Ramsden, Ian Pike, Mariana Brussoni.

**Data curation:** Rachel Ramsden.

**Formal analysis:** Rachel Ramsden.

**Funding acquisition:** Mariana Brussoni.

**Investigation:** Rachel Ramsden, Mariana Brussoni.

**Methodology:** Rachel Ramsden, Mariana Brussoni.

**Project administration:** Rachel Ramsden, Mariana Brussoni.

**Resources:** Rachel Ramsden, Mariana Brussoni.

**Software:** Rachel Ramsden.

**Supervision:** Ian Pike, Sally Thorne, Mariana Brussoni.

**Validation:** Rachel Ramsden, Mariana Brussoni.

**Visualization:** Rachel Ramsden.

**Writing – original draft:** Rachel Ramsden.

**Writing – review & editing:** Rachel Ramsden, Ian Pike, Sally Thorne, Mariana Brussoni.

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
