## [Decision Letter · Decision Letter 0]

18 Feb 2025

PONE-D-25-02992
Children’s outdoor play at early learning and child care centres: examining the impact of environmental play features on children's play behaviour
PLOS ONE

Dear Dr. Ramsden,

Thank you for submitting your manuscript to PLOS ONE. After careful consideration, we feel that it has merit but does not fully meet PLOS ONE’s publication criteria as it currently stands. Therefore, we invite you to submit a revised version of the manuscript that addresses the points raised during the review process.

We look forward to receiving your revised manuscript.

Kind regards,

Rajeev Singh

Academic Editor

PLOS ONE

Journal Requirements:

“This research was funded by the Lawson Foundation, grant GRT 2020-137.”

Reviewers' comments:

Reviewer's Responses to Questions

**Comments to the Author**

1. Is the manuscript technically sound, and do the data support the conclusions?

Reviewer #1: Partly

Reviewer #2: Yes

2. Has the statistical analysis been performed appropriately and rigorously? 

Reviewer #1: No

Reviewer #2: Yes

3. Have the authors made all data underlying the findings in their manuscript fully available?

Reviewer #1: Yes

Reviewer #2: Yes

4. Is the manuscript presented in an intelligible fashion and written in standard English?

Reviewer #1: Yes

Reviewer #2: Yes

5. Review Comments to the Author

Reviewer #1: Thank you for the opportunity to review this study. It reads well, but suffers from a lack of direction. The study questions are not teed up in the Introduction. The Intro should describe what is already known about associations between child play and the specific environmental features analyzed here so that by the time the reader reaches the measures and analytic strategy there are no surprises. Even in the Results, it is unclear exactly what question is being answered, and the second question seems to have disappeared completely. The Discussion is also a missed opportunity to highlight the novel insights provided by this study. I hope these comments allow the authors to significantly revise the manuscript to make it stronger.

Introduction

p. 5, line 107: “While there is existing evidence on outdoor play environments that support children’s outdoor play, many studies have identified a gap in the research on the effects of specific natural and built design features and how they are used.” What is that gap? What specific natural and built design features are referred to here? The next paragraph on diversity of play does not seem to resolve these questions.

The authors write: “…Gibson’s theory of affordances supports the important question, “Are children using these spaces as intended?”” But you go on to write that children may view and use spaces in ways designers may not foresee. Isn’t that contradictory? So why is it important to ask that question? If the answer is yes, what does that tell us? And what does an answer of no tell us? Also, the rest of the paper seems to ignore this research question.

By the time we get to “Research Purpose,” the study questions should be queued up in the reader’s mind, but they are not. Here, it says the first objective of the research is to analyze common environmental play features’ associations with children’s outdoor play. But what features exactly? What do we already know about those features’ associations with play? What outstanding questions does this study address? For example, are the descriptive statistics presented here on where play occurred novel or have other studies have conducted similar analysis? Just what does this study ask that others have not?

And the second objective presented here – understanding the alignment between children’s use and designers’ intentions – should be justified in light of the questions I raise in my previous paragraph. And again, I would ask the authors to describe what is already known about this subject and what outstanding question is addressed here and how.

Methods

Why were these 8 particular sites chosen?

How many coders were there? How were they certified as reliable?

Line 238: “…additional rules were determined to categorize play observations that were coded as non-play or restorative play and another play type.” What were these rules and what were the codes? Why does that matter if you’ve already dichotomized behaviors into play vs. non-play? What were the final play variables? Counts based across episodes?

Why were there 3 timepoints if the environmental play features didn’t change over time? Just to maximize the number of observations? If so, that’s fine, but say so. Specify which play features are considered fabricated and which are natural if this is an important distinction.

What is the “loose parts variable” mentioned in passing on line 254?

What is the purpose of showing Figure 2 and Figure 3? What are observational behavior points (line 256)? Figure 3 needs a legend.

If associations are being estimated between two things, they are bivariate, not univariate analyses (line 283).

It is odd to read that the purpose of a multivariate logistic regression model is to obtain an “adjusted total effect.” Isn’t the purpose to estimate independent associations between play features and play? But more importantly, why didn’t the Introduction prepare the reader for this line of inquiry?

The second research question presented in the Introduction seems to have disappeared.

How do models handle timepoint? How do they handle observations over time nested within centers? This is a critical oversight.

Why should centers with and without steep terrain be considered together, when they may have different associations between other environmental features and play? Same question for centers with and without loose parts.

Results

Percentages should be reported as integers.

Tables 5 and 6 do not need unadjusted ORs; that information was already presented in Tables 3 and 4.

The results regarding specific play types (Table 6) are too extensive for me to wade through. Because the Introduction did not prime me to attend to any particular dimension over any other, it is just too much information to process, which is why once again I return to the issue of what is important. If it is just a descriptive paper, all of the multivariate results are simply unnecessary. It may be sufficient to say, for example, that gardening areas are associated with children’s bio play without having to also claim that other types of features, much less unrelated factors like temperature, were controlled for. That may be a sufficient contribution to the literature, but I can’t tell based on the skimpy Introduction.

Discussion

The authors open this section by concluding that diverse outdoor play features offer more play participation for children than open areas. But all centers were examples of environments with diverse outdoor play features. Your coefficients only tell you about each specific feature relative to open area WITHIN a diverse outdoor environment. And was that the research question along?

The next paragraph makes it seem like that the number of types of play is the outcome of interest, but the models did not address that question (there were no formal tests across play features by number of types of play).

The authors should consider the possibility of making use of the variation among the 8 sites – rather than simply combining them all – to answer questions about how particular features work in the presence or absence of other features – again, if that is indeed their question.

By the end of this manuscript, I’m still not sure what specific knowledge gap this study addresses.

Reviewer #2: Thank you for submitting the manuscript titled "Children’s Outdoor Play at Early Learning and Child Care Centres: Examining the Impact of Environmental Play Features on Children’s Play Behaviour." The paper applies Gibson’s theory of affordances to explore the relationship between children’s outdoor play spaces and their behavioural patterns, which is a highly relevant topic. It effectively summarizes the current evidence and identifies an important gap in the existing knowledge.

Overall, I suggest updating the literature review. While many references are cited, I believe that including more current sources would strengthen the manuscript. For example, the latest special issue from AJOT (https://research.aota.org/ajot/issue/78/4) and other recent works could provide valuable insights.

While the use of observations and behavioural mapping is a child-friendly method, I would recommend considering a more holistic approach that centers on children’s own experiences and reflections. This would provide valuable insight into their reasoning and decision-making during play. Such an approach would complement the affordances theory proposed by Refshauge, which presents a more "able" perspective, yet also has its limitations. This theory is largely guided by an adult’s perspective on what constitutes, for example, "jump-on-able" or "balance-able" features. However, from a more dynamic standpoint, children with diverse abilities may identify and engage with these features differently. I believe it would be beneficial to mention children with varying abilities as key users of outdoor spaces to highlight the wide range of possible occupations and experiences they can have.

Additionally, I find that the manuscript is quite long, and some results are discussed in the discussion section but not explicitly presented in the results section. I suggest synthesizing and condensing the manuscript to make it more concise and focused.

Overall, I believe this is an interesting and important manuscript. The methodology employed is novel within this field and makes a valuable contribution to the existing body of knowledge.

6. PLOS authors have the option to publish the peer review history of their article (what does this mean?). If published, this will include your full peer review and any attached files.

Reviewer #1: No

Reviewer #2: No

---

## [Author Response · Author response to Decision Letter 1]

10 May 2025

Thank you for giving us an opportunity to revise and strengthen our paper. We have addressed all of the editorial and reviewer comments and questions in a detailed response to reviewers letter attached in this revised submission.

---

## [Editor Report · Decision Letter 1]

3 Sep 2025

Children’s outdoor play at early learning and child care centres: Examining the impact of environmental play features on children's play behaviour

PONE-D-25-02992R1

Dear Dr. Ramsden,

We’re pleased to inform you that your manuscript has been judged scientifically suitable for publication and will be formally accepted for publication once it meets all outstanding technical requirements.

Kind regards,

Rajeev Singh

Academic Editor

PLOS ONE
---

## [Editor Report · Acceptance letter]

PONE-D-25-02992R1

PLOS ONE

Dear Dr. Ramsden,

I'm pleased to inform you that your manuscript has been deemed suitable for publication in PLOS ONE. Congratulations! Your manuscript is now being handed over to our production team.

Kind regards,

on behalf of

Dr. Rajeev Singh

Academic Editor

PLOS ONE